# Near-future extreme sea level predictions from physics-informed deep networks

Anonymous Full Paper
Submission 6

## 001 Abstract

Storm surges, high-frequency extreme sea level events driven by the atmosphere, lead to loss of life and crucial infrastructure. Their prediction is a requirement to build adequate coastal protection, yet by design the current generation of coupled climate models cannot reliably do so. They have however long been used to predict and project changes in the atmosphere. We here re-train a recently produced physics-informed Long Short Term Memory (LSTM) network that was designed to reproduce observed extreme sea level events around Northern Europe. We notably reduce its number of predictors to the six atmospheric variables that the network ranked as most important and its temporal resolution from hourly to 3-hourly, so that we can pass as predictors climate model-produced atmospheric variables. We compare the climate model-based and observed sea level over 1985-2014 and select the most accurate runs to quantify changes in the number of extreme sea level events in the near-future (2025-2054) and end of century (2070-2099). The network projects on average increases in the number of extreme events, but there is a large spread in the predictions. In general, it also projects a larger increase in the near-future than at the end of the century. We attribute this spread to the inconsistent changes in the drivers: larger wind speeds, shifting from more west-northerly in the near-future to more west-southerly at the end of the century, yet higher sea level pressure. These results show the feasibility of predicting changes in extreme events from physics-informed deep learning networks, but more reliable predictors, from a wider range of climate models, are needed before the predictions can converge.

## 1 Introduction

Northern Europe is particularly vulnerable to extreme sea level events, as most of its population, financial and logistical hubs are located by the coast [1]. These events are expected to increase as climate change continues [2], but the exact numbers are uncertain. One main reason for this uncertainty is that our usual tool for climate change predictions and projections, global climate models, cannot -by design- reproduce accurate sea level [3]: Because notably of their vertical grid, the sea surface is often fixed and calculated afterwards, prognostically. They can however reproduce atmospheric processes [4], and we know that over Northern Europe, extreme sea level events are atmosphere-driven [2].

Using machine learning to predict extreme sea level events from atmospheric drivers is a logical next step. Using neural networks, Hieronymus et al. [5] could show that 36h forecasts for the Swedish coast were as accurate and much faster to produce than traditionally done with hydrodynamics models. Dubois et al. [6] extended their work and produced sea level predictions for the Baltic coast for the entire 21st century with Random Forest. More computationally demanding but more accurate, Barzandeh et al. [7] used an encoder-based deep network to predict extreme sea level along the coast of Estonia, while Heuzé et al. [8] used instead a Long Short Term Memory (LSTM) recurrent neural network, applying it to stations along the North Sea and Baltic coasts. These deep learning works focussed on reproducing and explaining observed sea level, rather than projecting long term changes.

Using transfer learning, we here re-train the LSTM network developed by Heuzé et al. [8] so that it can provide extreme sea level projections for stations around Northern Europe. Using an approach similar to that of Dubois et al. [6], we use as predictors outputs produced by the climate models that participated in the Climate Model Intercomparison Project phase 6 [CMIP6, 4]. After an assessment of the performances of this approach by comparing observed and climate model-inferred sea level over the historical period, we quantify changes in extreme sea level events for the near-future (2025 - 2054) and the end of the century (2070 - 2099).

## 2 Data and Methods

The Zenodo link and DOI of the scripts necessary to reproduce our results will be provided in the camera-ready version.

### 2.1 Sea level and atmospheric observational datasets

We use hourly tide gauge data from the same nine stations around Northern Europe as used by Heuzé et al. [8]: Den Helder (NL), Esbjerg (DK) and Lowestoft (UK) on the North Sea coast; Gedser (DK),

Helsinki (FI) and Umeå (SWE) on the Baltic coast; and Gothenburg (SWE), Malmö (SWE) and Oslo (NO) in the transition between the two seas. For the three Swedish cities Gothenburg, Malmö and Umeå, the tide gauge data were provided by the Swedish Meteorological and Hydrological Institute; for the other locations, we used the Global Extreme Sea Level Analysis (GESLA) dataset version 3, last updated in November 2021 [9, 10]. We de-tided the tide gauge data using the UTide package for Matlab [11], keeping the trends in.

The observed atmospheric variables used to train the network are from the ERA5 reanalysis [12], available at 0.25° resolution, hourly, since 1st January 1940. We use the hourly 10 m u- and v-components ($u_{10}$ and $v_{10}$) of the wind and the mean sea level pressure, at the grid cell closest to the tide gauge station coordinates as provided by SMHI or GESLA. We further computed the wind speed as $WS = \sqrt{u_{10}^2 + v_{10}^2}$, and split the wind components into their positive (i.e. westerly or southerly) and negative (i.e. easterly or northerly) components.

The highest temporal resolution available for CMIP6 models is 3-hourly; we therefore retimed all our hourly series, producing 3-h bin mean values with bins including times 0,1,2h, 3,4,5h etc, to be consistent with the CMIP6 times.

## 2.2 Output from the Climate Model Intercomparison Project phase 6

We use all CMIP6 models that at the time of download (June 2025) had the 3-hourly 10 m u- and v-components of the wind and the mean sea level pressure ('uas','vas', and 'psl' respectively in the CMIP nomenclature) for the historical run and at least one of the climate change runs. This amounted to exactly two models, from the same modelling family: MPI-ESM1-2-LR and MPI-ESM-1-2-HAM [13]. We acknowledge that this is not representative of the CMIP-model potential for such projection, and can only hope that future iterations of CMIP will feature more models with high temporal resolution output.

We chose the Shared Socioeconomic Pathway 3-7.0 as for these models, it was the one with the most ensemble members available [datasets 14–17]. In total, for both experiments, we have 50 ensemble members for MPI-ESM1-2-LR (51 for the historical run) and 3 for MPI-ESM-1-2-HAM; we use them all in the historical assessment, and limit our future projections to the 10 best performing of both models considered together. We assessed their performance by ranking the models based on their ability to reproduce the observed 99th percentile in sea level, for all nine cities, and taking the ten with the lowest combined rank.

For all climate models and ensemble members, for each city, we extracted the wind and pressure values of the grid cell closest to the city's tide gauge coordinates. As for the observations, we also compute the wind speed and decompose the wind components into their north/south/east/westerly contributions. We used those as predictors for our deep network, as described in the next section, to produce sea level values. We quantify biases in these re-created sea level values on the last 30 years of the historical run (1 January 1985 to 31 December 2014) and perform the sea level projections over the 30-year periods "near-future" (1 January 2025 - 31 December 2054) and "end of century" (1 January 2070 - 31 December 2099).

## 2.3 Long Short Term Memory and its re-training

We re-use the model architecture of Heuzé et al. [8], which they provided at https://doi.org/10.5281/zenodo.15754554. In brief, a deep network was trained to reproduce extreme sea level around Northern Europe from atmospheric predictors. This network is a Long Short Term Memory [LSTM, 18], which is a type of recurrent neural network that can take complex temporal dependencies into account. The LSTM, based on the Python package Keras [19], has 3 layers of 100 units each with a hyperbolic tangent activation function, and one dense layer, with a drop out rate of 0.015 in between each layer, an overall learning rate of 0.01, and 12h windows. For the original hourly predictors, the input batch size was 20; for our 3-hourly predictors, we use a batch size of 7.

We not only reduced the temporal resolution from hourly to 3-hourly, we also reduced the number of predictors to the six that were deemed most important for the region [8]: the 4 wind components, the wind speed, and the sea level pressure. We could directly re-use the same LSTM architecture and obtain great performances (Fig. 1): correlation larger than 0.9 with the observed sea level, and overall RMSE of 0.04, reduced to 0.02 or less for the high values (larger than 0.66 when normalised).

All datasets, be they from observations or CMIP6 models, are min-max normalised prior to feeding to the LSTM.

# 3 Results and Discussion

## 3.1 Historical assessment

We first use the 3-hourly CMIP6 outputs of sea level pressure and winds for 1985-2014 as predictors in our network, and compare their produced sea level to that from tide gauge observations over the same time period. We focus on the 99th percentile of the time series, which over 30 years of 3-hourly data yields 877 events.

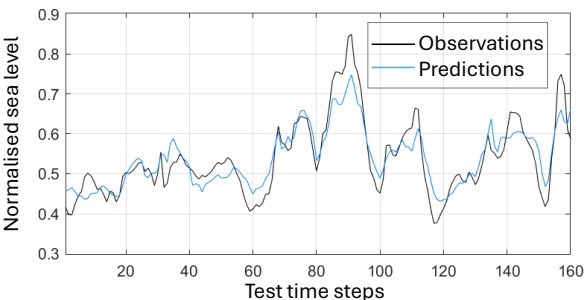

**Figure 1.** Performance of the LSTM prediction (blue) against its test set (black), for tide gauge observations from Gothenburg, roughly at the centre of the region of interest.

The performances of the best and worst ensemble members are shown for each city, for the two CMIP6 models, in Table 1. Starting with the best runs, we find that six of the locations have the correct value of their 99th percentile over the historical level, with biases in the number of events in that percentile lower than 1%. The other three cities have biases lower than 10% for MPI-ESM1-2-LR, worsening to nearly 14% for MPI-ESM-1-2-HAM, i.e. the observed highest 1% of the sea levels occur 15% of the time in the model. Performances are somewhat consistent across the ensemble members. That is, if the best run has a low bias, so does the worst run (Table 1): the six cities we highlighted previously reach at most 5% error for their worst run, while Den Helder, Esbjerg and Gothenburg have 5 to 18% bias. As expected, the range best-to-worst is smaller across the 3 ensemble members of MPI-ESM-1-2-HAM than across the 51 of MPI-ESM-1-2-LR (0.1-2.5% vs 1-10%).

Parts of these biases and their spread most likely stem from differences in the distribution of the drivers. Looking again at Gothenburg, which is in the centre of our region of interest and one of the cities with higher biases, we find that both the best and worst ensemble members have their mean sea level pressure shifted low compared to the reanalysis (Fig. 2a), with the shift more severe for the worst performing models. The modelled u-wind simultaneously has too few low (-5 to 0 m/s) and too many high (-15 to -5 m/s) easterly values, with the worst run also having too wide a distribution for the low westerly values (Fig. 2b). The modelled v-wind is centered on 0 m/s and has a symmetric distribution, while the observed has a southerly shift; if anything, the best performing run seems to be most biased. That is, predictor biases across ensemble members are inconsistent.

Interestingly, not one single ensemble member is best in all cities, even for MPI-ESM-1-2-HAM that has only 3 to choose from. For the climate change analysis, for each city, we therefore rank all ensemble

**Table 1.** Best and worst runs of the two CMIP6 models, i.e. minimum and maximum biases. For each city, First column: 99th percentile sea level in the observations and corresponding value when the series is min-max normalised; Second, minimum percentile difference, in %, corresponding to that value (0 if it is the 99th percentile for the model too, positive bias if that sea level value is more common in the model); Last column, maximum difference in % in percentile corresponding to that value. 'LR' and 'HAM' refer to the two CMIP6 models MPI-ESM1-2-LR and MPI-ESM-1-2-HAM, respectively. Second row is the run number of the corresponding ensemble member, in the CMIP6 format r[run number]i1p1f1.

| City | Obs 99 prct (min-max norm.) | | Best run LR | HAM | Worst run LR | HAM |
|---|---|---|---|---|---|---|
| DeH | 0.89 m | (0.59) | 8.8 | 13.9 | 18.1 | 16.5 |
|  |  |  | 46 | 3 | 31 | 1 |
| Esb | 1.22 m | (0.6) | 6.1 | 9.5 | 11.9 | 10.9 |
|  |  |  | 28 | 3 | 13 | 2 |
| Ged | 0.59 m | (0.71) | 0.0 | 0.0 | 1.5 | 0.5 |
|  |  |  | 11 | 1 | 13 | 2 |
| Got | 0.58 m | (0.62) | 2.6 | 3.6 | 7.3 | 4.8 |
|  |  |  | 33 | 2 | 30 | 1 |
| Hel | 0.62 m | (0.67) | 0.6 | 0.5 | 2.5 | 1.8 |
|  |  |  | 42 | 3 | 15 | 2 |
| Low | 0.68 m | (0.69) | -0.1 | 1.0 | 3.0 | 2.1 |
|  |  |  | 37 | 3 | 31 | 2 |
| Mal | 0.47 m | (0.68) | 0.5 | 1.2 | 3.8 | 2.3 |
|  |  |  | 22 | 1 | 20 | 2 |
| Osl | 0.65 m | (0.61) | -0.2 | 0.7 | 4.6 | 5.2 |
|  |  |  | 42 | 2 | 2 | 3 |
| Ume | 0.64 m | (0.71) | 0.1 | -0.3 | -0.8 | -0.4 |
|  |  |  | 15 | 1 | 14 | 3 |

members based on their bias, combining the two CMIP6 models, and then take their median rank across all nine cities. We ignore r2000i1p1f1, which is not available for the future runs. We take the 10 with the lowest median rank, which corresponds to an average rank strictly lower than 20 (out of 54), and analyse their predictions in the next section. These 10 all belong to MPI-ESM1-2-LR, and are (best first): r40, 18, 7, 22, 48, 12, 50, 11, 46, and 17i1p1f1.

## 3.2 Near-future and end-of-century extreme sea level events

The LSTM network predicts a large range of changes in sea level extremes even after selecting for the 10 best ensemble members (Table 2). On average, it predicts an increase in the number of events, except for Umeå where they will decrease. Five (four) of the cities have a median change stronger (lower) in the near future than by the end of the century. For all cities, except Umeå in the near-future, the most extreme predictions are of opposite sign. That is, despite a projected increase on average, there is at least one ensemble member that predicts a decrease. In

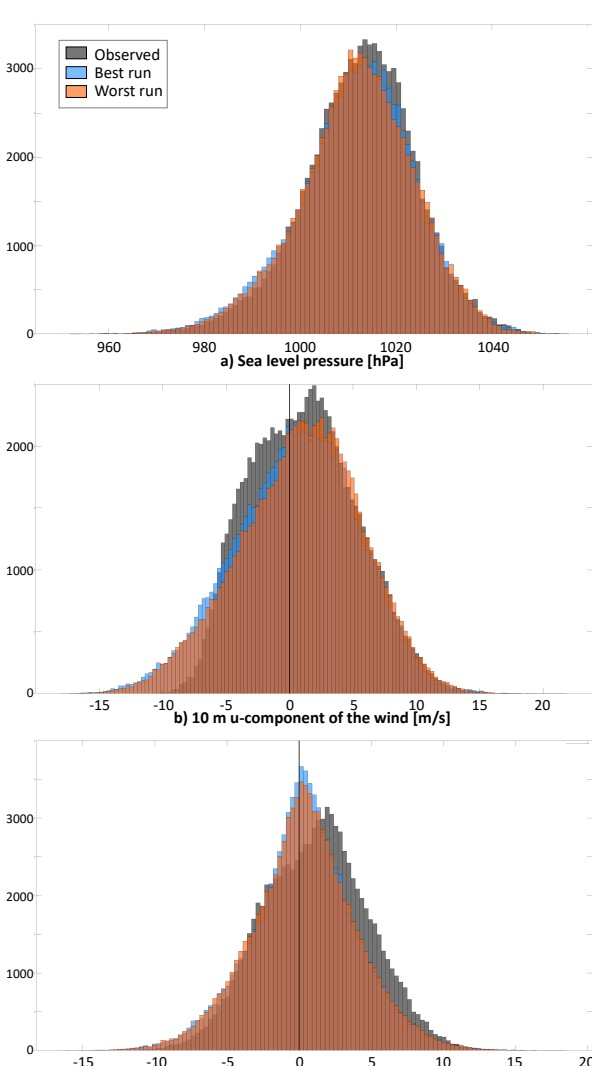

**Figure 2.** Distribution of the three 3-hourly atmospheric variables over 1985-2014 for Gothenburg, in observations (grey), in the best (blue) and worst (orange) performing ensemble member according to Table 1.

general, this projected possible decrease is stronger in the near-future than for the end of the century, but so is also the projected maximum increase. That is, as has been shown for other variables [e.g. 20], the internal variability becomes relatively weaker as the climate change forcing increases.

The reason for this range of predictions probably is the inconsistent changes in the atmospheric drivers (Fig. 3). The sea level pressure (a) is shifting towards higher values, i.e. less stormy conditions, which would be consistent with predictions of fewer extreme events. The winds in contrast are shifting away from 0. The u-component (b) is becoming more westerly, which over northern Europe means stormier conditions and therefore more extreme sea level events. The v-component shifts northerly, i.e. stormier, in the near-future (Fig. 3c, green), and then shifts southerly by the end of the century (purple). These are consistent with the stronger in-

**Table 2.** Change in the number of events with sea level higher than the observed 99th percentile (from Table 1) between the near future (2025-2054) and the end of century (2070-2099) compared to the historical period. Table shows the minimum, median and maximum across the 10 ensemble runs.

| City | 2025 - 2054 | | | 2070 - 2099 | | |
|------|-----|--------|------|------|--------|------|
|      | min | median | max  | min  | median | max  |
| DeH  | -7231 | 670  | 5554 | -4228 | 1662  | 4568 |
| Esb  | -2676 | 915  | 3888 | -390  | 1000  | 4570 |
| Ged  | -286  | 293  | 895  | -391  | 185   | 741  |
| Got  | -2500 | 1241 | 3333 | -2294 | 1063  | 2964 |
| Hel  | -1242 | 35   | 897  | -574  | 124   | 785  |
| Low  | -1356 | 8    | 1715 | -1202 | 210   | 1276 |
| Mal  | -293  | 713  | 1777 | -480  | 547   | 1174 |
| Osl  | -2393 | 1373 | 2116 | -2921 | 542   | 2617 |
| Ume  | -404  | -160 | -62  | -338  | -4    | 404  |

crease in the near-future. Earlier results [8] had also pointed out that Umeå was the one location where southerly winds were important predictors: this end-of-century southerly shift probably explains the emergence of predictions with increased extreme sea level events for that city (Table 2, bottom right cell).

Which atmospheric change is the correct one? The sea level pressure is less biased than the wind components (Fig. 2), so its projections of decreased storminess ought to be more trustworthy, but most literature on climate change anticipate more storms [21]. Ideally, one would have access to more CMIP6 models, from diverse modelling centres, and would select only those with accurate atmospheric drivers. Unfortunately only two models, from the same model family, released the 3-hourly output needed for this study. The upcoming CMIP7 [22] is putting a large emphasis on studying extreme events, so we can hope that more model output will become available soon for this type of work. Given that a recent study, using different CMIP output and Random Forest, found a similarly large range predictions [6], a better option may be to work with bias-corrected output, although correcting related output with opposite biases is not trivial.

Finally, it is important to bear in mind that this is only the atmosphere-induced part of extreme sea level events. In reality, the baseline of sea level is shifting [2], and waves and tides are changing too [23]. Even if the atmosphere were to become more stable and cause fewer extreme sea level events, these other processes will still increase the vulnerability of our coastlines [1].

## 4   Conclusion

We re-trained a Long Short Term Memory network that had been designed specifically to reproduce

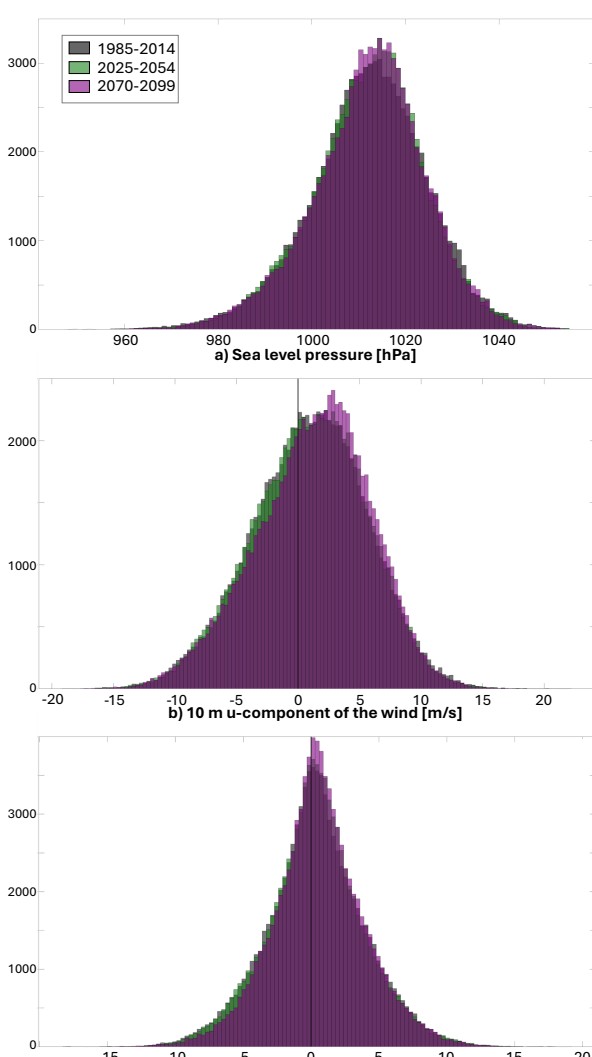

**Figure 3.** Distribution of the three 3-hourly atmospheric variables for Gothenburg for the historical period 1985-2014 (black), the near-future 2025-2054 (green), and the end of century 2070-2099 (purple).

extreme sea level around Northern Europe [8], so that it can now take as predictors climate model-produced atmospheric variables instead of the original observational datasets. That is, we reduced the number of predictors to the four wind components (northerly, southerly, westerly, easterly), wind speed, and sea level pressure, coming from three climate model-produced variables, and reduced the temporal resolution from hourly to the model-outputted 3-hourly. We used all models and all ensemble members that had these three variables available for the historical and climate change run SSP3-7.0, which amounted to only two models from the same model family and 54 ensemble members. We selected the 10 ensemble members that produced the most accurate historical extreme sea level (Table 1), and found an overall increase in the number of extreme sea level events in the future (Table 2) but with a large range of uncertainty. This is because the dif-

ferent atmospheric variables have large biases (Fig. 2) and surprising opposite changes (Fig. 3). As sea level keeps rising in response to sustained climate change, the risk of extreme events will only increase [2]. Such predictions are therefore urgently needed, and urgently need to become less uncertain, for example by increasing the number of models that release 3-hourly output [22].

# Acknowledgments

This work was funded by [anonymised] grant no. [anonymised] awarded to [anonymised]. We acknowledge the World Climate Research Programme, which, through its Working Group on Coupled Modelling, coordinated and promoted CMIP6. We thank the climate modeling groups for producing and making available their model output, the Earth System Grid Federation (ESGF) for archiving the data and providing access, and the multiple funding agencies that support CMIP6 and ESGF. We acknowledge the data access and computing support provided by [anonymised].

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
