# OpenReview forum: "Near-future extreme sea level predictions from physics-informed deep networks"
_NLDL.org/2026/Conference — Submitted to NLDL 2026_

### Official Review · Reviewer_cyHJ · 2025-09-18

**Rating:** 1
**Confidence:** 2

**Summary:**

This is an applied paper on sea level prediction, where an existing LSTM model is repurposed to use climate model variables as input.

The work largely skips the methodological part, and focuses on the analysis of the sea level predictions from that domain's perspective. As ML person it's difficult to position or contextualise the results: I don't really understand sea sciences.

**Strengths:**

The authors have used an earlier atmospheric model, which seems to be a suitable choice for this problem. The sea level prediction results are analysed quite comprehensively, and they seem convincing.

**Weaknesses:**

This is very applied work: it takes a black-box ML model, applies it to domain data, and analyses the results. I see little ML contribution or significance here.

The problem wasn't clearly defined. I'm not sure what sea level prediction specifically means, or what "extreme" sea level means. Furthermore, the sea level problem wasn't contextualised, or the contributions here positioned to the wider literature. There are no comparisons between different methods, data sources, or even ablations of different model tunings or choices. The paper should also demonstrate why this problem or method is significant or important.

The results look quite black box to me. It's not clear how the LSTM does the predictions, or how the predictions look like. What are the inputs and outputs to the LSTM? Not sure. The LSTM itself also seems a bit overkill: I think the model is using only six input variable, and surely some simpler model (eg. random forest) could work here well (or even better).

I also struggled to follow the result analysis. The paper does not clearly define what it means by "bias" or "spread", and I'm not sure what these mean. I'm not sure what "percentile difference" means. Maybe this is trivial to the authors, but I have not encountered this kind of error analysis before in ML.

Finally, the paper has no discussion about cross-validation or test/validation sets, and seems to hand-pick ensemble members. This all feels very risky: how do we now that the model is not overfitting or there is no data leakage? I find it dubious that the paper is choosing best ensemble members to fit all historical data, and then only tests on future data. Surely we should also test the performance on some historical data as well?

**Justification:**

This is an applied work which does not clearly demonstrate it's significance within an ML conference. The work also is poorly positioned to the wider literature, and the data analysis was difficult to follow. I'm a bit confused why this work was submitted to an ML venue in the first place: why should an ML audience learn about this?

---

> ### Author Rebuttal · Authors · 2025-10-15
>
> We appreciate the reviewer's honesty regarding their own confidence in their evaluation. We here address the main misunderstandings.
>
> 1) The detailed problem definition was covered in Heuzé et al. (2025), doi: 10.5194/os-21-1813-2025. In the interest of space, we only listed the main points in the introduction, along with relevant references. most of them to review papers. We therefore disagree that we were not "positioned to the wider literature" nor did we demonstrate that "the problem is significant or important".
>
> 2) We disagree with the reviewer's assessment that the model and results are "black box".
> We clearly refer to the work of Heuzé et al. (2025) and even included a link to their Zenodo repository in the Methods section. We indeed did not explicitly state that the main objective of their paper was explainability, on observations. That is, via permutation feature, they show which atmospheric variables are most important for sea level predictions in the various regions. That is the reason why we chose to retrain their model using only a subset of their many variables: from their deep learning work, we knew which variables were most important. We will make this clearer in the revised version.
> We also clearly describe all the changes that were done to the DL model and variables in section 2.3.
>
> 3) The reviewer is correct in that we assumed that the meaning of "bias" or "spread" would be obvious to anybody with a knowledge of statistics, which we assumed means everybody in the ML community. We will add a few words to explain these in the revised version.
>
> 4) We are not sure what the reviewer means regarding their final comment.
> On the one hand, no, in the interest of space, we did not re-explain the test/validation sets nor the overall training of the ML model, referring the interested reader to the work of Heuzé et al. (2025).
> On the other hand, the entire section 3.1 is dedicated to the evaluation of the ensemble members and why we picked these, and already tests "the performance on [the] historical data" as the reviewer seems to suggest we do.
> Either way, our point is that what the reviewer seems to want to see done has indeed been done.
>
> Finally, "why this work was submitted to an ML venue in the first place: why should an ML audience learn about this?"
> We disagree with the reviewer that there is such a thing as an homogeneous "ML audience".
> Our type of work is "applied deep learning". That is, we develop deep-learning based methods with the objective to address issues of relevance to science and society as a whole. We agree that this work may be of no interest to people whose work focusses on developing new methods within an IT performance framework, testing them against standard benchmarking sets.
> But from past experience of presenting this type of work at similar conferences, many in the audience were like us on the applied side, and many more on the non-applied side still want their ML work to be of use to society.
> Besides, as clarified in this rebuttal, a large part of the work in this paper is about ML-method development.
> We therefore argue that this paper fits perfectly within the scope of this conference.

---

### Official Review · Reviewer_JKX5 · 2025-09-28
**LSTM-based forecasting of extreme seal levels - Requires further improvement**

**Rating:** 1
**Confidence:** 4
**Final Rating:** 1
**Final Confidence:** 5

**Summary:**

The authors designed a model based on RNNs to forecast extreme seal levels. The model is a smaller derivation from the model in [1] with changes in time alignment and reduction in features. Results show a consistent behavior  with respect to historical data.

**Strengths:**

* The authors  are tackling a relevant problem associated with climate change and sustainable goals with relevant machine learning techniques.
* The project has strong foundations from [1] and indeed, the results in figure 1 suggest the proposed model has a strong potential to tackle the problem.

**Weaknesses:**

* Although the paper mentions there is an association to physics-informed deep networks, there is no explicit relation whatsoever to the topic. For instance, the proposed model is an LSTM trained on sensor data without explicit incorporation of physical laws. Thus the title and abstract are misleading. For instance, [1] acknowledge they work opens the doors for PIML, but the proposed model is purely data-driven.
* As the paper lies in the intersection between ML and geoscience, it is strongly recommended to discuss with more details the problem formulation and describe the proposed contributions in terms of machine learning vocabulary. Indeed, Sections 2.1 and 2.2 were difficult to follow for a reader who is not aware of [1].
* By reducing the number of models, the authors sacrificed the uncertainty quantification behind the predictions, as the prediction ensemble now relies on only 10 samples instead of 50. Indeed, there is no discussion of this uncertainty.
* The paper provides no information regarding the testing methodology with ground truth data. Are the results in Table 1 and Figure 1 derived from training data? If so, this raises concerning issues associated with the experimental design, as there is no tangible way to validate the models' performance. Although figures 1 to 3 suggest relevant results visually, there are no metrics that quantify this performance, e.g. RMSE, MAE, $R^2$. And as a follow-up comment, there is no discussion on computational usage, as one of the main points of the paper was to propose an efficient mode.
* The feature selection seems arbitrary, as there is no explicit justification for them.
* It is strongly recommended to expand on the proposed transfer learning methodology from the LSTM in [1]. How was the retraining process?
* No reproducibility information. The authors can share the code in an anonymized repository.
* Typo in the keywords and uninformative TL;DR.
* Finally, the model needs to be compared against other relevant baseline models for extreme earth events [2].

**Final Justification:**

Based on the discussion with the authors, it is noticeable the proposed work requires methodological refinement and further evaluation to accept the work. Some parts of the methodology are arbitrary and the comparison with respect to other methods or the state of the art is weak; finally, the use of certain words mislead the reader, e.g. physics-informed, transfer learning. Nevertheless, I encourage the authors to keep working on the problem, as it is highly relevant for AI for meteorology and natural disasters prevention.

**Justification:**

Although the paper addresses a relevant problem for the Scandinavian region using ML techniques, the proposed methodology is weak and misleading in terms of baselines and results. Thus, there is no obvious incremental contribution in the paper's current state. The manuscript requires further rewriting to improve clarity and presentation for the ML community.

Reference
[1] C. Heuzé, L. Carlstedt, L. Poropat, and H. Reese, “Drivers of high-frequency extreme sea levels around northern Europe – synergies between recurrent neural networks and random forest,” Ocean Sci., vol. 21, pp. 1813–1832, 2025, doi: 10.5194/os-21-1813-2025.
[2] S. Zhao, Z. Xiong, J. Zhao, and X. X. Zhu, “ExEBench: Benchmarking Foundation Models on Extreme Earth Events,” arXiv preprint arXiv:2505.08529, 2025.

---

> ### Author Rebuttal · Authors · 2025-10-16
>
> We thank the reviewers for their comments. We here address the points that they listed under potential weaknesses, by order of appearance:
>
> 1) The reviewer is right, "data-driven" would have been a better term to use. We will modify this in the revised version of the manuscript.
>
> 2) In the interest of space, we limited how much of [1] we included in our text, but we could indeed have written more and will be happy to do so when we submit a revised version;
>
> 3) We had a long discussion with the co-authors while designing the study regarding this point. The main conclusion from section 3.1 "historical assessment" is that all ensemble members are strongly biased and the spread between these biases is low compared to the bias itself. That is, as others as shown before, the bias of individual ensemble members is not the result of internal variability but rather indicates that the dynamics are wrong in the CMIP models. We therefore decided to wrap this short study using only the 10 "least bad" ensemble members, and are now conducting the full study on bias-corrected variables. Expanding on these results is however beyond the scope of this paper.
>
> 4) The reviewer seems to have missed the performance metrics of section 2.3. We could indeed have expanded on the methodology, but again we chose not to repeat too much of [1] in the interest of space. We will write more about it in the revised version.
> As for the computational usage, for reasons on which I cannot explain without risking de-anonymising this text, everything had to run on the main author's nothing-fancy laptop. Once the re-training was complete it took barely 2-3s per 150 year time series to run. We do not remember exactly how long the re-training took, but not long at all.
>
> 5) Again, the feature selection is explicitly addressed in section 2.3.
>
> 6) Same response regarding the transfer learning methodology (see section 2.3), but we agree that this could have been expanded. We will expand in the revised version.
>
> 7) Our funding agency requires that we use Github/Zenodo, which to the best of our knowledge would have broken the anonymity criterion.
>
> 8) The last keyword seems to have been cut short indeed. Suggestions for better TL;DR would have been appreciated.
>
> 9) We are funded to provide projections of extreme sea level around Scandinavia, primarily using long-term records from tide gauges (i.e. 1D input). As interesting as [2] seems, their flood events do not include wider Europe and are based on SAR, i.e. are 2D images that cover only the recent past. If our model performs well for other extreme events of [2] that is great but that is not our mandate, so we leave it to other people to test.

---

### Official Review · Reviewer_FCHA · 2025-10-09
**Good process, Significant discussion, Uncertain results, Great work!**

**Rating:** 4
**Confidence:** 4

**Summary:**

The authors train an ensemble of small LSTM models (< 1K weights) to predict the sea level at 9 locations in Northern Europe based on ERA5 reanalysis wind and sea level pressure using historical observational meteorological data as labels. The models are validated on observational data and compared to CMIP6 data for comparison to physics based models and forecasts for the near future (2025-2054) and end-of-century (2070-2099) are computed and analyzed. The results indicate that extreme sea level events correlated with wind and sea level pressure could become more common in the near-future with a slight moderation towards the end of the century. There is a significant amount of uncertainty in the result, but this is clearly stated and reflected upon.

The climate analysis contribution is bigger than the machine learning specific contribution. At the same time, the work displays a good use an LSTM model and it's predictive performance and a strong analysis section that should inspire to better use of innovative ML frameworks to make a bigger impact in the climate domain without relinquishing physical or statistical rigor.


## Questions
### Q1
If I understood correctly, you chosen to split the wind data into 5 components, why? In particular, the north-south and east-west components are perfectly (negatively) correlated and thus provides the model with the same information.

### Q1
In the comparison with CMIP6-data, you chose to show the best and worst of the selected ensemble members. This is a rather non-traditional way of representing spread of model predictions. It does provide some insight, but also (as you mention) gives the impression that more ensemble members increases the spread of the results even for a static distribution. Do you have an impression of the skewness of the distribution of ensemble members for the 99th percentile level?

### Q3
Based on Table 1, it seems the difference in bias from best to worst model could be bigger for the locations where the model bias is higher, i.e. that the spread and predictive performance are correlated. Do you observe this?

**Strengths:**

The work is clearly motivated and the analysis and discussion are significant and well considered. In my opinion, this is a reasonable analysis process for using deep learning for climate forecasting in general: 1: Use of very high quality labels for training and testing of an ensemble model (observational data at the locations where the measurements were taken). 2: Comparison to physics based simulation data (with reflection on the source of deviations) and 3. Forecast analysis using the ensemble, not only single runs.
Validation of deep learning models is generally difficult with such flexible models and the authors of this work have clearly considered this and make significant efforts to both validate the general framework overall, but more specifically to validate and understand the particular data samples of interest, i.e. the 99th percentile events. At the same time, the authors do not overstate the effectiveness or robustness of the trained model but make clear the role of uncertainty.

**Weaknesses:**

There are several data sources used in this work; ERA5 reanalysis data (partially simulated, partially observational), CMIP6 (simulated, but physically coherent) and finally the extremely high quality meteorological observational data of sea level from the particular locations. For those less familiar with Climate datasets, it could be stated more clearly of what data quality each data source should be considered especially as context to better understand the comparison to the CMIP6 data.

**Justification:**

The work is clearly motivated and the analysis and discussion are significant and well considered. The evaluation focuses on the parts of the data predictions that matter most and the physical component is left out of the predictive model when using an LSTM is considered well by comparing to CMIP6 data. The results are not overstated, but reasonably discussed.

---

> ### Author Rebuttal · Authors · 2025-10-16
>
> We thank the reviewer for giving us a score of 4. Despite this score, to respect their time, we answer their three questions and address the one mild weakness they listed:
>
> Q1: This is to be consistent with Heuzé et al. (2025) who first built the model. They justify it with the min-max normalisation: it is not straightforward to min-max normalise a variable that can be either positive (northward/eastward) or negative (southward/westward). Besides, they de-trended the variables as well, which presents the same challenges when the variable can take both signs.
>
> Q2: It is rather common in the bias assessment literature, in order to show "how bad" the CMIP models can be. However, as has been pointed out before in the literature, the spread is rather small compared to the biases themselves, i.e. the issue is not internal variability but rather wrong dynamics. Ideally we would have worked with the best CMIP models rather than just the best ensemble members of one model, but there was no other model to choose from because of our constraint on the temporal frequency.
>
> Q3: Yes, we do. As the rest of the project will focus on one location only, we however decided to not investigate this further.
>
> And we agree with the reviewer that we could have better described each of the data sources. In the interest of space, we simply provided references and let the interest reader go and check them. We will add some text in the revised version.

---

### Official Review · Reviewer_qXR7 · 2025-10-10
**Review for Submission 6**

**Rating:** 4
**Confidence:** 3

**Summary:**

This work adapts an existing LSTM network to accept 3-hourly climate model outputs for predicting extreme sea level events in Northern Europe. It projects increases in extreme events for 2025-2054 and 2070-2099 under SSP3-7.0, though with large uncertainty due to limited model availability and inconsistent atmospheric driver changes.

**Strengths:**

1. The authors successfully demonstrate transfer learning by adapting an hourly-trained LSTM to 3-hourly climate model data. This reduces predictors from many to 6 atmospheric variables while maintaining good performance.

2. The paper includes a historical assessment (1985-2014) comparing model-predicted sea levels against observations, with detailed bias analysis across ensemble members and cities. The authors provide transparency about model performance before making future projections.

3. The authors acknowledge the limitation of having only two CMIP6 models from the same modeling family, transparently reporting the large uncertainty ranges in projections, and avoiding overconfident claims.

**Weaknesses:**

1. The authros reduced it to 6 predictors and 3-hourly resolution based on historical importance, but provide no independent validation that these choices work for future climate states. The LSTM should be validated on a split historical period to confirm it generalizes beyond its training conditions.

2. Only 2 CMIP6 models from the same modeling family provide the required 3-hourly data, with 53 ensemble members representing only initial condition uncertainty, not structural model uncertainty. This produces projections spanning opposite signs for 8/9 cities, which indicates insufficient model diversity rather than meaningful climate uncertainty. Without comparing against structurally different models, you cannot distinguish whether these ranges reflect true climate uncertainty.

3. The results show contradictory atmospheric changes. The sea level pressure increases while westerly winds strengthen, with v-winds shifting direction (Fig. 3, lines 272-284). But the authors do not validate whether these patterns are physically plausible by comparing them to other CMIP6 models. If the two MPI models produce contradictory atmospheric drivers, the LSTM propagates these inconsistencies into unreliable sea level projections without adding physical understanding.

**Justification:**

This paper demonstrates a methodologically sound transfer learning approach to adapt an LSTM network for climate model inputs. Despite limited CMIP6 data availability (only 2 models) that prevent reliable projections, the work provides insights into data requirements for future climate model intercomparison projects.

---

> ### Author Rebuttal · Authors · 2025-10-16
>
> We thank the reviewer for giving us a score of 4. Despite this score, as a sign of respect for the reviewer's time, we reply to the three points they listed as potential weaknesses:
>
> 1) The leading hypothesis not only of this work but of our overall funded projects is that the drivers of past extreme sea levels will continue being the main drivers of extreme sea levels in the future. Rather than testing on different observational periods, we direct the interested readers to this recent review by Melet et al. (2024), doi: 10.5194/sp-3-slre1-4-2024, which highlights that there is a consensus in the scientific literature that this will be the case around Northern Europe, which is already cited in the manuscript. We will add a sentence to the revised version to make this point clearer.
>
> 2) We fully agree, as we point out in the Methods, Results, Discussion, and Conclusion. Unfortunately, as we explain in the text, these two CMIP models are the only ones that outputted their 3-hourly surface wind and sea level pressure variables.
>
> 3) Again, we agree and wish there were more CMIP models to choose from. But there are none.

---

### Meta-Review · Area_Chair_j1c8 · 2025-11-04

**Recommendation:** Reject
**Confidence:** 4

**Metareview:**

dear authors,
thanks for your initial submission and the effort made in the rebuttal phase. As pointed out by reviewers, the work shows both positive and negative aspects. Despite the positive ones (including the clear motivation, the analysis and discussion generally significant, and the insights, and the partial soundness of the methodology), I feel the shortcomings of the paper are too strong to consider acceptance of the paper in NLDL 2026. Indeed, some methodological and evaluation flaws remain, and the experiments are very limited, especially the comparison with the state of the art.
In this context I recommend to reject the paper.

---

### Decision · Program_Chairs · 2025-11-05

**Decision:**

Reject

**Comment:**

Based on the reviewers and AC comments, the paper cannot be presented at the conference.